# The Comparative Effects of Mesenchymal Stem Cell Transplantation Therapy for Spinal Cord Injury in Humans and Animal Models: A Systematic Review and Meta-Analysis

**DOI:** 10.3390/biology10030230

**Published:** 2021-03-16

**Authors:** Louis D. V. Johnson, Mark R. Pickard, William E. B. Johnson

**Affiliations:** 1Chester Medical School, University of Chester, Chester CH1 4BJ, UK; 2University Centre Shrewsbury, University of Chester, Shrewsbury SY3 8HQ, UK; m.pickard@chester.ac.uk

**Keywords:** mesenchymal stem cell, spinal cord injury, ASIA score, AIS grade, BBB score, functional outcome

## Abstract

**Simple Summary:**

The effects of mesenchymal stem cell (MSC) transplants on functional recovery after spinal cord injury have been compared in humans and animal models. Data show that MSC transplants increase functional outcomes across species. However, modes and timings of MSC delivery mean that the animal studies cannot be used to predict outcome, suggesting that alternative models are required to improve translation of research to clinical trial.

**Abstract:**

Animal models have been used in preclinical research to examine potential new treatments for spinal cord injury (SCI), including mesenchymal stem cell (MSC) transplantation. MSC transplants have been studied in early human trials. Whether the animal models represent the human studies is unclear. This systematic review and meta-analysis has examined the effects of MSC transplants in human and animal studies. Following searches of PubMed, Clinical Trials and the Cochrane Library, published papers were screened, and data were extracted and analysed. MSC transplantation was associated with significantly improved motor and sensory function in humans, and significantly increased locomotor function in animals. However, there are discrepancies between the studies of human participants and animal models, including timing of MSC transplant post-injury and source of MSCs. Additionally, difficulty in the comparison of functional outcome measures across species limits the predictive nature of the animal research. These findings have been summarised, and recommendations for further research are discussed to better enable the translation of animal models to MSC-based human clinical therapy.

## 1. Introduction

Spinal cord injury (SCI) is a destructive event that can occur in humans and other animals. The symptoms depend on the severity of the injury and include loss of motor and sensory function and a decreased life expectancy, with SCI survivors frequently dependent on medical resources and social support [1]. An estimated 250,000 to 500,000 people suffer a SCI every year worldwide, the majority of which are from driving and sporting accidents [2]. There are few effective treatments for SCI and none that restore function in chronic paraplegics. Consequently, researchers are exploring new molecular and cellular investigative options. For these prospective treatments, an animal model is often used in a preclinical setting to establish safety and potential efficacy. However, the genetic and physical differences between species can be large, prompting the key question as to how predictive preclinical results in animal studies are of what is likely to happen in a human clinical setting. This systematic review and meta-analysis are aimed to gather information and data from the available literature to allow comparison between the results seen in animal models of SCI and human SCI in a qualitative and, where appropriate, quantitative manner. While researchers are exploring many prospective treatments for SCI, the review has focussed solely on the surgical transplantation of mesenchymal stem cells (MSCs), which represent one of the more commonly explored cell therapies for SCI [3].

### 1.1. Spinal Cord Injury

Traumatic SCI involves primary damage to the spinal cord from mechanical impact, which disrupts and destroys local neuronal axons and blood vessels [4]. Following this initial damage at the site of impact there is a cascade of secondary damage to surrounding tissues. Disruption of microvasculature causes haemorrhage and subsequent haemostasis, which blocks off the blood supply, ultimately leading to tissue ischemia, cell necrosis, and subsequent inflammation, with an influx of immune cells [5]. Excitotoxic levels of extracellular glutamate released as a result of cell necrosis contribute to further cell death and axonal injury [6], and microglial cells become activated and upregulate pro-inflammatory cytokines, such as tumour necrosis factor (TNF) α and interleukin (IL)-1 β [5]. Astrocytes that are peripheral to the epicentre of the injury proliferate and become hypertrophic, which, over time, produces a large mesh-like network of cellular processes contributing to formation of a glial scar [7]. This string of events begins in what has been termed the immediate phase (≤2 h post SCI), before continuing into the early acute (≤48 h post SCI) and sub-acute (≤2 weeks post SCI) phases of SCI [8]. The time period from 2 weeks to 6 months, bridging the gap between the sub-acute and chronic phases, is known as the intermediate phase of SCI. Within this phase continued maturation of the glial scar occurs, with the formation of a densely layered chondroitin sulphate proteoglycans (CSPGs) that act as a physical and biological barrier to axonal regrowth, contributing to the inhibition of any motor or sensory functional restoration [9]. In addition to this, the death of oligodendrocytes leads to demyelination of axons and reduced conduction of action potential, contributing to a further loss of function [10]. There is some evidence of remyelination within the intermediate phase; however, the regenerated myelin sheath is often abnormally thin and does not restore normal action potential conductance [11]. The chronic phase of SCI (>6 months post SCI) is characterised by stabilisation of the lesion site, full development of the glial scar around fluid filled cysts, and a plateau in any functional recovery [6,12].

In humans with a chronic SCI, there is no natural improvement in motor or sensory function [6]. Hence, treating chronic SCI could be considered the most difficult and least effective option, as the secondary damage cascade has fully realised its effects. However, deciding whether a SCI is acute, sub-acute, or chronic can be somewhat arbitrary. While the chronic phase has been described by Rowland et al. [6] as the time beyond 6 months after injury, others would categorise a SCI as chronic after 1 year, and only if there was a plateau in functional recovery in the preceding 6 months [13]. This discrepancy in when an SCI is classified as sub-acute or chronic has been evidenced in this systematic review and meta-analysis, where Karamouzian et al. [14] described patients with an SCI of between 2 weeks and 6 months post injury as sub-acute, while Hur et al. [15] described patients of 3 months post SCI as chronic. Evidently, there is an element of subjectivity and variation involved in the categorisation of SCI phases.

### 1.2. Current Treatments and the Need for New Therapies

Treatment options for SCI are very limited. Early surgical decompression of the spinal cord is often used to remove any compressive tissue, e.g., vertebra and to stabilise the lesion, but also to alleviate neuropathic pain and identify potential problems that may occur in the future, such as syringomyelia [16]. Pharmaceuticals, such as paracetamol and non-steroidal anti-inflammatory drugs (NSAID), can be given, the effects of which also involve reducing inflammation and pain-relief. Methylprednisolone (MP) is an anti-inflammatory corticosteroid shown to promote survival of endogenous cells in animal models of SCI [17]. MP has been used clinically to treat SCI patients; however, a recent systematic review and meta-analysis reported that MP treatment showed a lack of significant long-term benefits and increased risk of side effects, such as gastrointestinal bleeding [18]. In addition, to date there is little evidence of MP treatment having beneficial effect when administered later than the early stages of the acute phase of SCI [19]. Hence, there is still a pressing need for safe and effective medical interventions both to prevent the secondary damage that occurs in the acute and sub-acute phases after SCI and also with restorative potential in the longer-term and chronic patients.

In recent years, researchers have pursued several new cellular therapy and molecular approaches to treat SCI. In animal models, transplants of induced pluripotent stem cells [20], neural stem/progenitor cells (NSPCs) [21], olfactory ensheathing cells (OECs) [22], and MSCs [3,23,24] were shown to promote functional recovery. The putative mechanisms of action for these cell therapies vary from neuronal cell replacement strategies to trophic support that enhance axonal regeneration and need further investigation. For example, MSCs secrete neuroregulatory factors, such as nerve growth factor (NGF) and brain-derived neurotrophic factor (BDNF), which promote neural growth, along with angiogenic factors, such as vascular endothelial growth factor (VEGF) [24,25,26,27]. This trophic nature of MSCs is likely to be important in SCI repair mechanisms. MSCs were reported to have anti-inflammatory activity that decreased the presence of inflammatory cytokines and reduced neuronal cell death when transplanted in the early phases of injury [23], whilst also preventing the development of neuropathic pain [28]. In addition, MSCs have been shown to promote nerve growth within nerve-inhibitory CSPG rich microenvironments, both in vitro [29] and in vivo [30]. An alternative approach to cell transplantation to enhance nerve growth has been the use of chondroitinase ABC (ChABC), which is an enzyme that breaks down the dense nerve-inhibitory CSPGs within the glial scar that block axonal regeneration [31]. Research that has shown further that a combination of interventions may enhance axonal growth and neurological recovery; hence, ChABC has been used combined with cell transplantation therapies, including transplanted Schwann cell [32], NSPCs [33], OECs [34], as well as MSCs [35].

The rationale for many of these approaches is that it will be necessary to change the microenvironment in the spinal cord to reduce damaging inflammatory activity, decrease the presence of scar tissues, whilst promoting the potential of surviving or replacement neurons to regenerate and restore neuronal function. However, to date, it is unclear whether such experimental approaches will successfully translate to an effective clinical treatment procedure for SCI in humans.

### 1.3. In Vivo Models of Spinal Cord Injury

In vivo models of SCI are undoubtedly important, as they allow examination of the safety and efficacy of investigative interventions, which cannot be determined in vitro. When attempting to replicate the conditions of human SCI, researchers have used a range of animal models including rat, mouse, rabbit, dog, cat, pig, and non-human primate [36,37,38,39,40,41,42]. Each animal model has its benefits. Small rodents such as rats and mice are cost effective and have relative ease of husbandry [43]. Rats, more so than mice, are also advantageous in that their SCI pathophysiology is more akin to that of humans with the formation of a glial scar and fluid-filled cysts, which is not seen in mice [44,45]. The intrinsic capacity for recovery in rats and humans is also more comparable than in mice and humans, in that any functional improvement plateaus in both species, although rats commonly have improved recovery in the acute-sub acute phases [45,46,47]. For these reasons, the rat model could be considered superior to the mouse model in terms of how well it can be used to represent human SCI. However, there are still important differences between rodent models, including rats, and humans. These include variations in the signalling inputs of the corticospinal tracts to motor neurons in the spinal cord, disparity in recovery time and extent between rats and humans, which could be due to the differences in scale of lesions and requirements for the length of axonal growth to affect neural functioning [43,44,45,46,47].

In contrast, the dog SCI model is valuable due to a closer size and etiology to humans. Dogs, like humans, can be subject to naturally occurring SCI [48]. The main cause of this is intervertebral disc (IVD) herniation; a degenerative disease caused by compression of the spinal cord by a herniated IVD, preceding the secondary cascade of destructive events previously described. Additionally, an estimated 7% of SCI cases in dogs are due to traumatic events such as road accidents [48]. Because of this, it is possible to conduct studies on both experimental dogs with a purposefully induced SCI [49], and companion (pet) dogs that have suffered accidental spinal trauma [50], which is most commonly the causation of human SCI.

Perhaps the closest animal model to humans with regards to similarity of the central nervous system (CNS) structure/function relationships, scale, SCI pathology, and behavioural change, is the non-human primate model [47]; the most frequently of which studied is the monkey [43]. Experimenting on monkeys, however, is costly and requires specialised long-term care [43]. In addition, inflicting SCI on non-human primates, although carrying major benefits in terms of developing a better understanding of future human SCI treatments [47], nonetheless warrants intense ethical consideration and concern [51,52]. Thus, there is far less research performed using these models, including research with MSC transplants, as evidenced from the lack of data emerging from literature searches performed within this systematic review.

### 1.4. Study Objective

This study was performed to examine any similarities or differences between SCI functional outcomes in animals and humans after MSC transplantation and hence to help determine the extent to which the results seen in the animal models are representative, or even predictive, of those seen in humans. This is an important comparison and question as animal studies often are performed as a preclinical measure towards the development of future human therapy. However, functional recovery measures differed between the species examined, which meant it was not possible to compare outcomes in a quantitative manner. Additionally, there were very few randomised, controlled trials (RCTs) of MSC transplants in humans, which contrasted with a greater number of controlled studies in animal (mostly rat) models. Therefore, both a qualitative and a semi-quantitative approach were taken to compare the functional outcomes of MSC transplants across species. Meta-analyses were completed for the available data in human studies to examine changes in outcome over time, and in those controlled animal studies that presented appropriate data as a way of examining the efficacy of MSC treatment within animal species. We report on these comparative findings and further discuss issues associated with the translation of the animal models to human clinical MSC applications.

## 2. Materials and Methods

### 2.1. Literature Searches and Study Selection

Studies were selected after completing searches on the following databases: PubMed, ClinicalTrials.gov (accessed on 30 June 2020), and the Cochrane Library. The search was performed on terms used for all search engines were “mesenchymal stem cell” AND “spinal cord injury”, plus, for PubMed only, a separate search on “mesenchymal stem cell” AND “spinal cord injury” NOT review, plus a separate search on “mesenchymal stem cell” AND “spinal cord injury” AND “canine”. In PubMed, the following filters were applied: “Clinical trials, within the last 10 years”, “Case study, within the last 10 years”, “Other animals, within the last 10 years”. In ClinicalTrials.gov, the search was filtered for “Complete only studies”. No filters were applied to the Cochrane Library search. Searches were performed up until 30 April 2020. The identified studies were then screened according to inclusion and exclusion criteria, as detailed.

### 2.2. Inclusion and Exclusion Criteria

The inclusion criteria were as follows: studies that reported functional outcomes of an MSC transplant in the treatment of SCI in either human clinical trials, including case or pilot studies, non-randomised or uncontrolled studies, phase I–III trials; studies that reported on functional outcomes in animal models of SCI, including non-randomised, uncontrolled and controlled studies. Some animal studies included multiple cohorts, i.e., animals were treated with MSC transplants as different times post SCI, in the acute, sub-acute, or chronic phase. These data were analysed as separate studies according to the SCI phase of treatment.

The exclusion criteria were as follows: cross-species treatments; tissue sources other than adipose, bone marrow and umbilical cord; adjunct treatments; MSCs co-cultured with other cell types; in vitro research; MSCs administered any way other than intrathecally, via lumbar puncture, or intravenously, as these methods only were used to treat human participants (note we have reported the delivery of cells as being via lumbar puncture or by intrathecal delivery, as originally described in each of the included studies, although it may be likely that all intrathecal injections were in fact by lumbar puncture, i.e., by intrathecal injection into the cerebral spinal fluid in the lumbar region); any studies that involved MSC modification, e.g., through cell surface engineering or viral transduction, except for fluorescence tagging for tracing purposes, which were included; studies without locomotor function outcomes; any studies where the research reported did not include primary data. Data were salvaged from research involving modified MSCs if the study treatment groups included MSC alone; these data were included only for those non-modified MSC groups. These inclusion and exclusion criteria were initially applied to the title and abstracts of identified studies, then to the full manuscript texts after this initially screening exercise (step 3 and step 4, respectively, of Figure 1).

### 2.3. Participants

Human clinical studies and trials included participants of all ages, genders, and initial American Spinal Injury Association (ASIA) scores. Animal models included all species that were identified in the available literature, i.e., rat, mouse, dog and rabbit, and included animals with naturally occurring SCI or surgically induced SCI, whether the injury was a contusion, compression or transection.

### 2.4. Outcome Measures

For human studies, the American Spinal Injury Association (ASIA) Impairment Scale (AIS) was used to measure functional outcomes of SCI, which is a precise measure to gauge severity of the injury and has become the most widely adopted measure of SCI outcome worldwide [53]. Patients are clinically assessed through a series of function-based tests and given an AIS grade ranging from ASIA A, which denotes a complete SCI, i.e., where there is no sensory or motor function below the sacral levels S4–S5 of the spine, to ASIA E, where an SCI has occurred, but there is normal motor and sensory function [54]. The AIS grades are themselves determined by ASIA motor scores and sensory scores, the latter consisting of light touch and pin prick scores. Upper and lower extremity muscle groups are given a motor score of between 0–5, where a score of 3 and above represents full range of motion against gravity. For a patient to receive an AIS grade of ASIA D, at least half of the muscle group scores below the neurological level must be at least 3 [54].

For animal studies, motor function was mainly assessed by the Basso–Beattie–Bresnahan locomotor rating scale (BBB score), ranging from 0, where there is no observable hindlimb movement, to 21, where the animal has coordinated gait, consistent toe clearance and trunk stability, with the tail consistently up [55]. The BBB score was used in all rat, rabbit and dog studies, apart from one dog study, which used a canine-specific adaptation of the BBB score, ranging from 0 to 19 [56]. In mouse studies, the Basso Mouse Scale (BMS) was used. The BMS ranges from 0, where the animal has no ankle movement, to 9, where the animal has consistent plantar stepping and normal trunk stability [57].

For the purposes of this study, for human studies treatments with MSCs within the first 48 h of SCI were considered acute, treatments from 48 h to 14 days post-SCI were considered sub-acute, treatments from 14 days to 6 months post-SCI were considered intermediate, whilst treatments after 6 months were considered chronic, using the phases of SCI proposed by Rowland et al. [6]. In the animal studies, MSC treatments within 48 h of SCI were considered acute, 48 h to 7 days were considered sub-acute, 7 days to 4 weeks were considered intermediate, whilst treatments at or after 4 weeks post-SCI were considered chronic. These phases were decided because the glial scar is fully matured in rats at around 3 weeks [44] and previous rat studies indicate a plateau in functional recovery at around 4/5 weeks [58,59].

### 2.5. Statistical Analysis

Meta-analysis to demonstrate efficacy of MSC therapy in human, rat and mouse studies was completed using SPSS software and Review Manager 5.4 software provided by the Cochrane Collaboration (London, UK) [60]. For the data on the human studies, the ASIA motor and sensory scores from each study were pooled and Shapiro–Wilk tests for normality were performed. As these data were not normally distributed, Mann–Whitney U tests were used to determine significance between groups, i.e., before and after MSC treatment, where *p* values below 0.05 were considered significant. For the data on animal studies, the BBB/BMS scores were inputted into Review Manager 5.4 software with means, SD values, and sample sizes. Standardised mean difference (SMD) was calculated using the generated forest plots with a 95% confidence interval. To assess heterogeneity, the inconsistency index was used (I2). If an I2 value of >50% indicated significant heterogeneity, then a random effect model was used for data analysis. If the I2 value was <50%, then a fixed model was used. The use of the same analytical tools and statistical techniques were also possible for the ASIA motor scores in the three human studies with a treatment group and a control group, where data were provided and could be extracted. Summary data have been shown as box and whisker plots for non-normally distributed data and forest plots showing standardised mean differences (SMDs), where heterogeneity was determined by Chi^2^ and I^2^ tests and *z* tests used to determine overall effect size. A funnel plot of SMD plotted against standard error was used to examine publication bias, which only was possible for the rat studies due to insufficient numbers of studies for the other species and subgroups analysed [60].

## 3. Results

### 3.1. Study Selection

After literature searches and screening of identified primary research papers, a total of 47 studies were included for the systematic review, with 19 human studies and 28 animal studies (Figure 1; see Table 1 and Table 2 for the human and animal studies, respectively).

### 3.2. Summary of Baseline Characteristics of Human and Animal Studies

The summary baseline characteristics of the participants and their treatments are shown in Figure 2.

The total number of human participants was 224. Of the human studies, 11 studies used intrathecal administration for MSC delivery, one study used intravenous administration for MSC delivery, two studies used a mixture of both intrathecal and intravenous administration for MSC delivery, and five studies used lumbar puncture as a way of delivering MSCs. Fourteen human studies used autologous bone marrow-derived MSCs (BM-MSCs) for transplantation, three human studies used autologous adipose tissue-derived MSCs (AT-MSCs), one human study used autologous umbilical cord-derived MSCs (UC-MSCs), and one human study used allogeneic UC-MSCs. Hence, 18 of the 19 human studies used autologous cells.

Of the human studies, 83% of the participants (*n* = 186) were in the chronic phase of SCI, whilst 17% were in the intermediate phase, whilst no participants were in the acute or sub-acute SCI phases. Of those with chronic SCI, 56% (*n* = 105) were classified with an AIS grade of ASIA A, 37% (*n* = 69) as ASIA B, 5% (*n* = 9) were ASIA C, and 2% (*n* = 3) were ASIA D. The remaining 17% of participants (*n* = 38) had an intermediate SCI, of which 95% (*n* = 36) were classified as ASIA A and 5% (*n* = 2) were ASIA B.

The total number of participants in all animal studies was 359. Of the 27 animal studies, 19 studies used a rat model, four studies used a mouse model, three studies used a dog model, and one study used a rabbit model. Moreover, 17 of the animal studies administered the MSC transplant intrathecally and 10 of the studies used intravenous administration for MSC delivery. Twenty-two of the animal studies used allogeneic BM-MSCs, the remaining five used allogeneic AT-MSCs. None of the animal studies used autologous MSCs. All experimental animals were submitted to an SCI that resulted in complete loss of function according to the BBB score, i.e., a score of 0 on the scale of 0–21.

Within the animal studies, approximately half of the animals included in the studies (*n* = 184) were categorised as acute when they were treated with MSC transplantation, 34% (*n* = 121) were classified as sub-acute, 1% (*n* = 4) were classified as intermediate, and 14% (*n* = 50) were classified as being within the chronic phase of SCI.

### 3.3. The Effects of MSC Transplantation on Functional Outcomes in Humans

Of the 38 patients with an intermediate SCI, 87% (*n* = 33) did not improve AIS grade, while the remaining 13% (*n* = 5) made a recovery from ASIA A to ASIA C. Of the 186 patients with a chronic SCI, 77% (*n* = 143) did not improve AIS grade, 14% (*n* = 26) improved from ASIA A to ASIA B, 3% (*n* = 6) improved from ASIA A to ASIA C, 5% (*n* = 10) improved from ASIA B to ASIA C, 1% (*n* = 1) improved from ASIA C to ASIA D. These results have been summarised in Figure 3.

From the literature that included ASIA motor and sensory scores, these data were extracted tested for statistical significance in those scores prior to and after MSC transplantation. Shapiro–Wilk tests showed that that ASIA motor score (*p* < 0.001), light touch score (*p* < 0.001), and pin prick score (*p* < 0.001), before and after treatment were not normally distributed. Therefore, Mann–Whitney U tests were performed to test for statistical significance between scores. It was found that there was a significant increase in all three scores following MSC treatment. In addition, three of the human SCI studies included data on the ASIA motor scores, but not sensory scores, in both the MSC transplantation group and control groups. This enabled a meta-analysis using forest plots, which demonstrated that in these controlled studies MSC transplantation was associated with a significant increase in motor score compared to control (Figure 4).

### 3.4. The Effects of MSC Transplantation on Functional Outcomes in Animals

Forest plots showing SMD were used, where data were available, to observe the efficacy of MSC transplants in animal models. There was only one RCT presenting mean and standard deviation values for the rabbit and dog studies; therefore, these models were not included in this analysis. Furthermore, any rat and mouse studies that did not report mean and standard deviations for functional outcome data were excluded.

Forest plots were performed separately for MSC transplants in acute, sub-acute and chronic SCI rat studies, for all rat studies, and for all mouse studies. These have been shown in Figure 5 and Figure 6. As shown, there was a significant improvement in functional outcomes in all of these experimental models. A funnel plot was performed to examine risk of publication bias for all of the rat studies examined, which has been shown in Figure 7. This analysis suggested that there may have been publication bias in favour of the intervention, i.e., MSC transplantation due to some evident asymmetry; however, the heterogeneity seen in the acute phase rats in particular would suggest that a formal conclusion of publication bias is not warranted.

Given there was only a single study using a rabbit model of SCI, efficacy could not be determined beyond that reported in the study. Lin et al. [38] reported that transplantation with allogeneic bone marrow-derived MSCs improved the locomotor function of rabbits to a mean BBB score of 12.5 while the control group improved to a mean score of 8.4 at 21 days post-treatment, with MSCs transplanted 1 day after SCI.

Within the three dog studies, locomotor function was measured using the BBB score (0–21 scales) in two studies [49,93], while the third used a canine-specific variation of this scale (cBBB) ranging from 0–19 [94]. These studies were small in sample size, with just four MSC-treated dogs in each. Lee et al. [93] conducted a controlled trial in which the MSC transplantation group had a mean BBB score of 6.8 vs. a BBB score of 2.5 in the control group (*n* = 4 dogs) at 8 weeks post-treatment, which was at 3 weeks after SCI. The study by Khan et al. [49] reported that MSC-treated dogs improved minimally to a mean score of 4.25 from 0, at 4 weeks post-treatment with MSC transplantation immediately post SCI, whilst Lee et al. [94] reported an increase in cBBB score from 0 to 7.5 within 8 weeks of treatment, with MSCs transplanted 1 week post-SCI.

There were no clear relationships between the MSC dosages, or if cells had prior treatment with cryopreservant (10% DMSO) with any of the safety or functional outcome measures in the human or animal studies.

## 4. Discussion

Animal models of diseases are useful because they enable researchers to observe safety and examine mechanisms of action of novel therapeutic interventions. In addition, they give an insight in to potential efficacy in human trials. Systematic reviews and meta-analyses are important methods to observe efficacy of new treatments [95]. A previous meta-analysis by Oliveri et al. [4] has shown that MSC therapy has some effectiveness in restoring motor function after SCI in rats. Since then, MSCs have been used in additional human trials. Furthermore, to date, no studies have directly examined how MSC transplants in animal models compare with the human clinical studies performed; hence, an up-to-date review that examines and compares the effects of MSC transplantation between animal and human studies is necessary to better understand how well the animal models may predict results in human therapies.

This systematic review and meta-analysis has focussed on reporting measures of functional outcomes. Although safety is an essential consideration in the human studies, it was not specifically examined or reported for the animal studies. The overall safety of MSC transplantation in humans with SCI was reported in most of the human clinical trials within this review, although mild adverse events (AEs) in the form of headaches, fever, and transient myalgia were not uncommon [66,67]. MSCs have been considered to be a safe stem cell therapy over the years, as transplants have not resulted in tumour formation [96,97]. However, the fact that MSCs can cause substantial AEs, such as hyperthermia and “fleeting” malignant hypertension [71] warrants further examination. Further, one of the human pilot studies in this review reported that an autologous MSC transplant could have long term detrimental effects after a participant suffered increased neuropathic pain on both sides of the body 12 months post-treatment [74].

Although the majority of animal studies did not report any AEs of any after MSC transplantation, this is not to say that the animals did not have AEs that may be similar to those reported in the human studies. It is probable the animal research studies did not investigate the possibility of AEs. For example, although Ban et al. [78] acknowledged that MSCs have been proven safe, i.e., no tumour formation in previous studies, in the introductory paragraphs, they do not mention safety or AEs thereafter. The same can be said for the animal studies reported by Chen et al. [79], Hosseini et al. [81], and Karaoz et al. [82].

Conversely, Watanabe et al. [28] reported on neuropathic pain in a rat model of SCI and have shown that MSC transplants reduced pain signalling and functional outcomes related to pain. Neuropathic pain is prevalent in human SCI patients [98,99] and the management of this of fundamental importance. Therefore, determining pain outcomes as well as other measures of AEs related to MSC transplants can and should be targets in future preclinical trials in order to gain more reliable and robust knowledge of the safety of MSC-based therapy for SCI.

### 4.1. The Efficacy of MSC Transplantation on Function Outcomes

Efficacy of MSC treatment in humans was assessed by improvements in locomotor and sensory function determined by changes in the AIS grade. While some patients improved an entire grade, (21.5% of all participants who were treated with MSC transplantation), the majority of the human participants did not (78.5% of all treated participants remained at the same baseline AIS grade throughout). The majority of these human studies did not have a control arm, which limits any firm conclusions that can be made regarding MSC transplantation therapy. However, those RCTs that were performed in humans have provided some promising results. The RCT of El-Kheir et al. [70] showed that 17/50 of MSC-treated participants improved an AIS grade while 0/20 participants in the control group improved an AIS grade. While the authors of this study acknowledged that the statistical analysis was underpowered, its findings show promise that MSC treatment may be superior to none at all. Similar results were seen in the RCT by Dai et al. [66], in which 9/20 of the MSC-treated group improved an AIS grade, while none of the 20 patients in the control group did. The RCT in humans performed by Cheng et al. [65] did not report any changes in AIS grade in either the MSC treatment group (*n* = 10 participants) or the control (*n* = 24 participants). However, this study did report that the MSC treatment group had significantly reduced muscle tension, increased limb strength, and increased bladder capacity/control than the control group, which was shown by an increase in the ASIA motor score and urodynamic analysis. These are still promising signs that MSC transplantation therapy can have a positive effect on overall quality of life for SCI patients. Other encouraging results reported after MSC transplantation include improved sexual function and anal sphincter contractions, though these effects also were seen in uncontrolled and non-randomised clinical studies [15,64].

With regards to improvements in ASIA motor and sensory scores, there was a statistically significant increase in all three scores in patients following MSC treatment. However, these data comprised only patients who underwent MSC therapy and has not been compared against data of control groups. Therefore, further research with randomised participant allocation into treatment groups and control groups are required, where the participants and clinicians involved in the assessment of functional outcomes are blinded to the participant allocation and treatment are required before it can be concluded that MSC treatment is beneficial. Nonetheless, the fact that MSC transplantation was associated with a significant benefit in some aspects of functional outcomes, and that these patients were largely in the chronic phase of SCI, is promising.

There were a relatively large number of studies of rat SCI models with an MSC transplantation group and a control group, which meant that a meta-analysis of the rat studies using available mean values and standard deviations could be performed. The forest plots of this analysis demonstrated that MSC transplantation therapy is favoured over non-treatment control groups. The mean BBB score of MSC-treated rats in all studies was 10.8, while the mean BBB score for the control groups was 6.9. A BBB score of 10–11 denotes occasional/frequent weight supported plantar steps and no forelimb–hindlimb coordination [55]. There was considerable heterogeneity (I2 > 50%) in the meta-analysis data for the acute and all rat studies, as opposed to little heterogeneity (I2 < 50%) in the meta-analysis data for the sub-acute and chronic rat models [60]. This may suggest that the outcomes of MSC transplantation in the later stages of SCI in rats are likely to become more predictable than in earlier stages post SCI.

In the mouse studies, the forest plot also supports that MSC transplantation therapy is favoured over control groups, because this was from only three studies, and there was a high heterogeneity (I2 > 50%), further research is required to confirm this conclusion.

The single study of MSC transplantation in rabbits provided evidence that the treatment is beneficial in rabbits. However, the rabbits used in the study had a mean BBB score of at least 3 within 24 h post-SCI and prior to MSC transplantation, meaning the injury was not complete prior to treatment, as it had been in the other animal models. Therefore, more studies using rabbits with a complete SCI are required to examine their use as a model of efficacy of MSC treatment for humans with ASIA A grade SCI. Similarly, further research is required using dogs as a model of SCI, as only one study had a control group to compare the effects of MSC transplantation [93]. Nonetheless, the data reported in all the canine studies may suggest that MSC transplantation is beneficial.

### 4.2. The Relevance of SCI Animal Models to Human Studies

To compare the efficacy of MSC transplants for SCI between human and animal models is difficult. In this study, it was decided to compare only those animal models that aligned with the modes of cell delivery performed in humans, i.e., using lumbar puncture/intrathecal and/or intravenous injections. Many animal models of potentially therapeutic cell transplants for SCI are performed whereby cells are delivered directly into the spinal cord itself, e.g., by transplanting cells using a gelfoam carrier system [24] or injections [100]. However, such an intervention itself may likely result in additional CNS damage and deleterious inflammatory responses, hence, clinical and basic researchers have sought alternative approaches [100,101]. Clearly, another important issue in comparing preclinical animal studies and human clinical studies is that there was a discrepancy in the length of time between SCI and the intervention of MSC transplantation, and also the fact that nearly all of the human studies were with autologous cells, whilst all of the animal studies were with allogeneic cells. The participants in the human studies were entirely in the intermediate or chronic phases of SCI, while the great majority of the animal studies were in the acute and sub-acute phases of SCI. This clearly raises questions concerning as to how well the animal models of MSC transplantation therapies may represent would be the likely outcome in the human clinical setting. As described, there are many pathophysiological changes that occur during the later phases of SCI, which may be detrimental to the effectiveness of MSC therapy, including the formation of the inhibitory glial scar [7]. It is understandable that most human cases of MSC transplantation have been within the later phases of SCI, due to the immediate need to maximise conservative approaches to patient care following SCI, the fact there is a lack of prognostic biomarkers in the early time points after injury, the time required to culture autologous MSCs for administration and the advantages of an autologous approach rather than using allogeneic cells, potentially requiring immunosuppressive drug administration [3,102,103,104]. In regard to this last point, it should be noted that in most of the animal studies, the immune status of the recipients was not clarified, whilst four rodent studies stated that no immunosuppressants [83,84,85,88] were administered on cell transplantation and one rodent study detailed the use of cyclosporine A [23]. Given that an important potential mechanism of action of MSC transplantation is their anti-inflammatory and immunomodulatory activity in the earlier stages of SCI [23,28], this area of research requires further investigation. These important disparities in treatment protocols should be taken into account when modelling SCI in non-human species, such that the conditions in the animal models are as close as possible to those of the patients being treated in clinic. There is evidence to suggest that some early interventions after SCI, including surgical decompression, but also potentially stem cell therapy, are more effective when undertaken in the acute phase [34]. In the interest of ensuring that animal models of MSC transplantation are more representative of human clinical studies, preclinical research should consider focusing more on treatments for chronic SCI.

Another reason for the difficulties in accurately comparing the functional outcomes in the human and animal studies is the differences in the methods and scales used to measure locomotion between species. The AIS grade used for humans may be considered more complex than the BBB score used in animal studies. The AIS grade ranges from ASIA A to ASIA E, which is ostensibly 5 grades, but in fact consists of multiple components consisting of different muscle groups and measures of sensation, while the BBB score has 21 grades that are based solely on the movement of animals. The subjective nature of aspects of these assessments and varying precision of these scales makes a direct quantitative comparison of recovery from humans to another species impractical. For example, a human patient who becomes weight-bearing would be given a grade of ASIA D, once they have an ASIA motor score of at least 3, i.e., with full range of motion against gravity in muscle groups below the neurological level [54]. However, this grade does not specify to what extent the patient is weight-bearing in the same way the BBB score does, wherein scores of 9 onwards (to a maximum of 21) specify the extent in which the animal can walk. To elucidate this in humans, ASIA motor and sensory scores would need to be considered; however, increases in these scores alone do not always represent an observable and meaningful functional benefit comparable to an increase in a whole AIS grade [54]. There are currently no set margins within the AIS system agreed upon that determine clinical importance, but a number of researchers have stated that a whole AIS grade is both significant and clinically desirable [105,106]. It remains illogical to attempt to identify which number on the BBB scale is equivalent to an ASIA D grade; this difference in scoring systems remains an important limitation in being able to extrapolate data from rat models to human studies.

### 4.3. Electrophysiology

As discussed, there are factors that limit the accuracy in predicting results from animal models of SCI to human trials. Hence, there is a need for a better way to compare the outcomes of investigative treatments between species. Some researchers have investigated electrophysiology as a way of measuring SCI lesion severity and nerve function. Following physical stimuli, such as light touch, vibratory, and proprioceptive sensations, somatosensory evoked potentials (SSEPs) can be monitored, which give an indication as to how intact the dorsal columns in the spinal cord are [107,108]. The monitoring of SSEPs is advantageous in that it is applicable in different species. Metz et al. [109] found that humans and rats have an analogous relationship with regards to locomotor function and electrophysiological outcomes in SCI, i.e., a shorter latency and higher amplitude of SSEPs correlated significantly with a higher BBB score in rats and higher ASIA motor scores in humans. Considering SSEPs can be monitored in both species, then these data suggest their measurement would substantially help bridge the gap in translational studies. However, the work of Mendonca et al. [61] on MSC transplantation in humans with SCI suggests that SSEPs and locomotor function are not always correlated. While 7/12 of the participants improved an entire AIS grade representing a significant functional improvement, only 1/12 of the same participants showed an improvement in SSEPs. Conversely, in the study of MSC transplantation in humans reported by Vaquero et al. [62], while seven participants had improved SSEPs, just four participants increased an AIS grade after MSC treatment. This evidence suggests that while electrophysiology investigation is useful for examining severity of SCI lesions, it cannot consistently predict locomotor function recovery.

### 4.4. Biomarkers of Functional Recovery

In more recent times, researchers have begun to observe correlations between routine blood analyte levels and neurological recovery of SCI patients. Brown et al. [104] conducted a preliminary study that measured the levels of 30 blood analytes, using principal component analysis (PCA) to group analytes into functions. It was found that the “liver function” analytes (alkaline phosphatase, alanine transaminase, and gamma-glutamyl transferase), and the “acute inflammation and liver function” analytes (C-reactive protein and total bilirubin) constituted the most prognostic value to their predictive model, suggestive that liver function was indicative of SCI recovery. Additionally, it was found that levels of these analytes in the blood may be affected by factors such as age, gender, smoking, existing health conditions, and certain medications [104]. Therefore, the use of blood analytes as biomarkers of neurological recovery are yet to be fully confirmed because, as was acknowledged by the authors, the sample size was not large enough to find significant correlation between specific analyte levels and ASIA motor and sensory scores at 3 and 12 months follow up. Tong et al. [110] found that blood albumin levels were a significant independent biomarker of functional SCI recovery. Taken together these points show how important it is to avoid confounding factors when examining the significance of a variety of biomarkers, such that correct conclusions can be drawn. Nonetheless, although future work must be undertaken with a larger sample size, there is potential for an inter-species method of predicting neurological improvement following SCI through measuring biomarkers that are seen in humans and, for example, rats and dogs.

## 5. Conclusions

SCI is a complex condition and finding an effective, restorative treatment is the ambitious focus for many researchers. Cell therapies, including MSC transplants, are trialed on experimental animal models to assert safety and investigate efficacy. This systematic review and meta-analysis has highlighted limitations of the animal models that negatively impact on how well they represent human SCI. The major limitations identified include an absence of rigour in the observation of animal safety, a disconnect in the phases of SCI being examined between species, i.e., mostly chronic SCI in humans and mostly acute/sub-acute SCI in animals, a difference in the cell types being administered, i.e., autologous MSCs in humans vs. allogeneic MSC in animals, as well as a difficulty in comparison between functional outcomes measures across species. Future work using animal models may be improved by increasing their safety observations, assessing changes in neuropathic pain, blood pressure, and temperature to be able to better understand the possibility of AEs in humans. It would also be beneficial for future animal trials to use a chronic model of SCI to better represent the human patients who are most likely to be treated. However, in addition to identifying underlying limitations in current animal research used to model human SCI treatments, this review has shown that MSC transplantation in animal models and humans is beneficial, resulting in significant improvements in ASIA motor and sensory scores, although the clinical importance of increases in these functional outcomes in humans is unclear and cannot be concluded due to a lack of control groups in most studies. Further research involving appropriately controlled human trials with larger participant cohorts, along with better aligned animal studies, potentially incorporating electrophysiology and biomarkers of functional recovery, will both enable better comparison of results across the species, and also help bridge gaps between pre-clinical research and the development of successful new clinical therapies.

## Figures and Tables

**Figure 1 biology-10-00230-f001:**
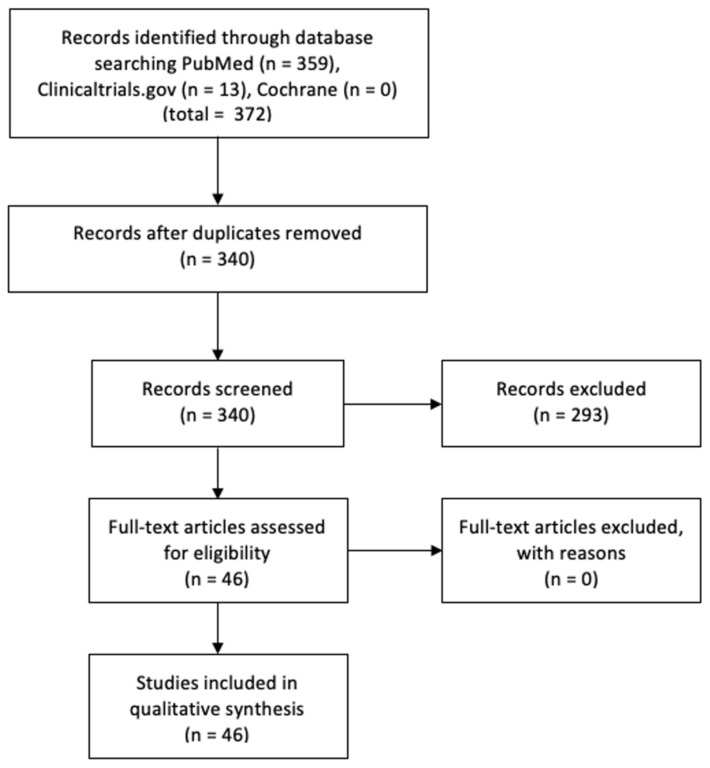
Preferred Reporting Items for Systematic Reviews and Meta-Analyses (PRISMA) style flow diagram for inclusion and exclusion of studies.

**Figure 2 biology-10-00230-f002:**
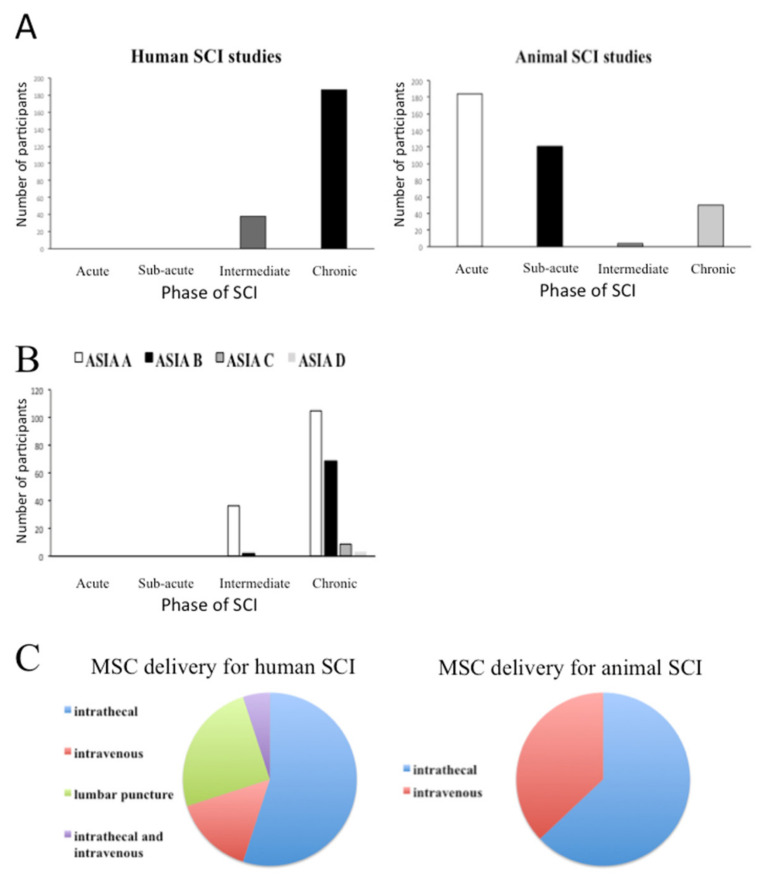
Summary of base characteristics of included human and animal studies. (**A**) Overall numbers of human and animal participants treated with MSC transplantation in different phases of SCI. (**B**) The baseline AIS classifications of human SCI participants prior to MSC transplantation. (**C**) The modes of MSC delivery in the human and animal studies.

**Figure 3 biology-10-00230-f003:**
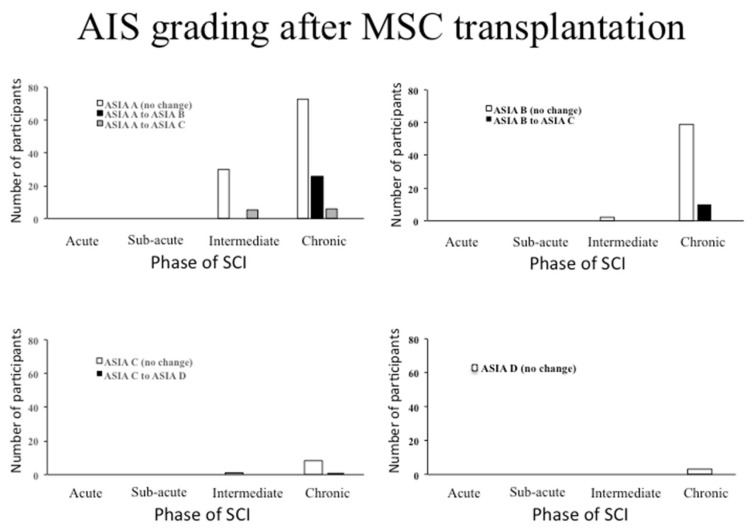
Change in AIS grading in human studies of MSC transplantation. The changes in a whole AIS grade have been shown for each phase of SCI when the participants were treated. Top left panel: participants with a baseline AIS grade of American Spinal Injury Association (ASIA) A and their recorded improvement. Top right panel: participants with a baseline AIS grade of ASIA B and their recorded improvement. Bottom left panel: participants with a baseline AIS grade of ASIA C and their recorded improvement. Bottom right panel: no participants with a baseline AIS grade of ASIS D improved to ASIA E (normal motor and sensory function).

**Figure 4 biology-10-00230-f004:**
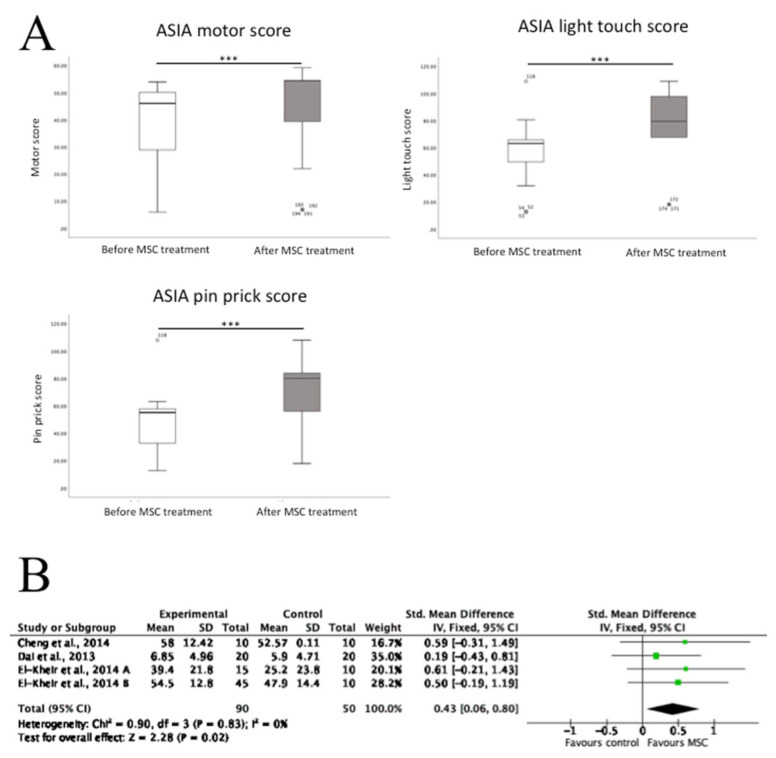
The effects of MSC transplantation on motor and sensory function in human SCI participants. (**A**) There were significant increases in ASIA motor scores, ASIA light touch scores and ASIA pin prick scores following MSC transplantation in all human participants who had been administered with cells. Data shown as box and whisker plots from 10 studies for ASIA motor score (*n* = 129 patients) and nine studies for ASIA light touch/pin prick scores (*n* = 119 patients). *** *p* < 0.001, Mann–Whitney U tests (**B**). In those human studies with a control arm, there was a significant increase in ASIA motor scores in the MSC treatment group compared with the control group. A forest plot is shown of standardised mean differences; SD, standard deviation; IV, inverse variance; CI, confidence interval. Heterogeneity was determined by Chi^2^ and I^2^ tests, whilst a *z* test determined overall effect size.

**Figure 5 biology-10-00230-f005:**
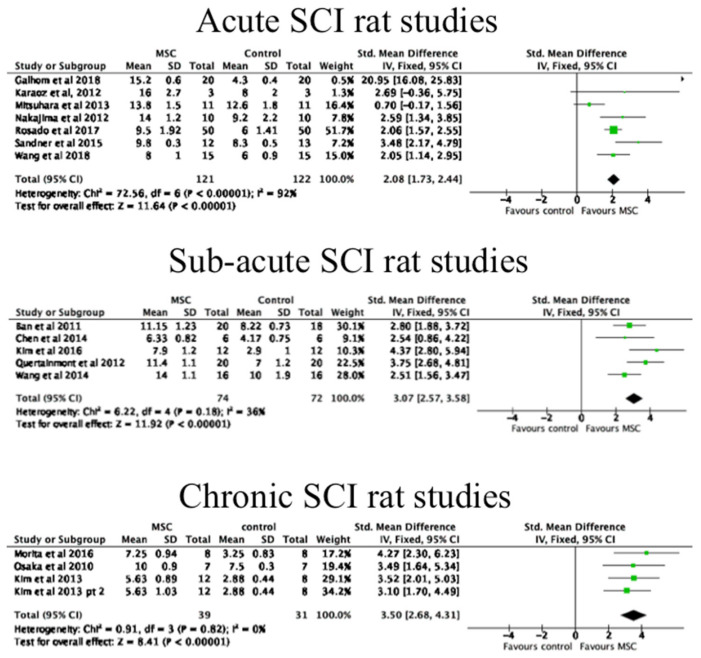
The efficacy of MSC transplantation in rats at different phases of SCI. As shown, there were significant increases in functional outcome in each phase of SCI. Forest plots are shown of standardised mean differences; SD, standard deviation; IV, inverse variance; CI, confidence interval. Heterogeneity was determined by Chi^2^ and I^2^ tests, whilst a *z* test determined overall effect size.

**Figure 6 biology-10-00230-f006:**
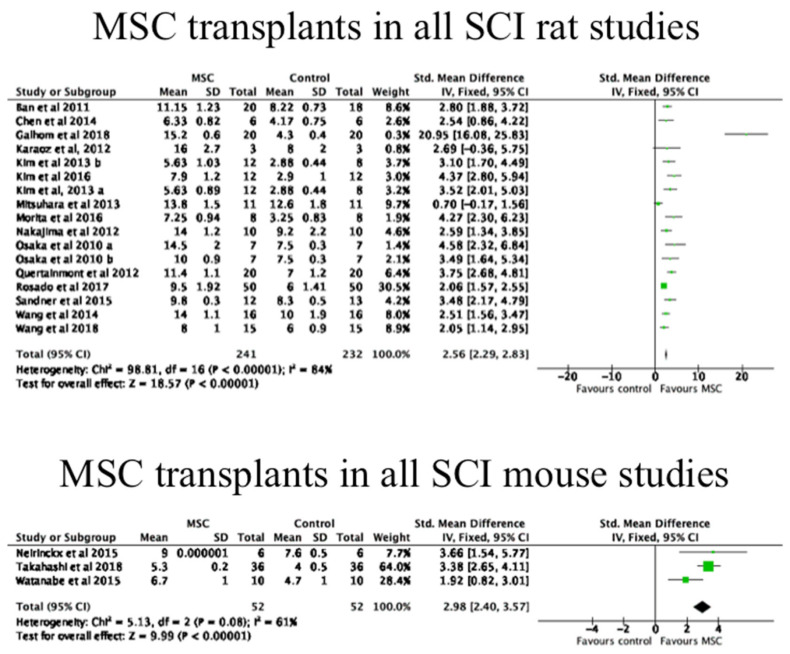
The efficacy of MSC transplantation in all rats and mice with SCI. Data shown as forest plots for extracted data from MSC transplant studies in all rats and mice. As shown, there were significant increases in functional outcome in both species. The mouse studies included one study with an MSC transplants in the acute phase [92] and two studies with MSC transplants in the sub-acute phase [28,37]. Forest plots are shown of standardised mean differences; SD, standard deviation; IV, inverse variance; CI, confidence interval. Heterogeneity was determined by Chi^2^ and I^2^ tests, whilst a *z* test determined overall effect size.

**Figure 7 biology-10-00230-f007:**
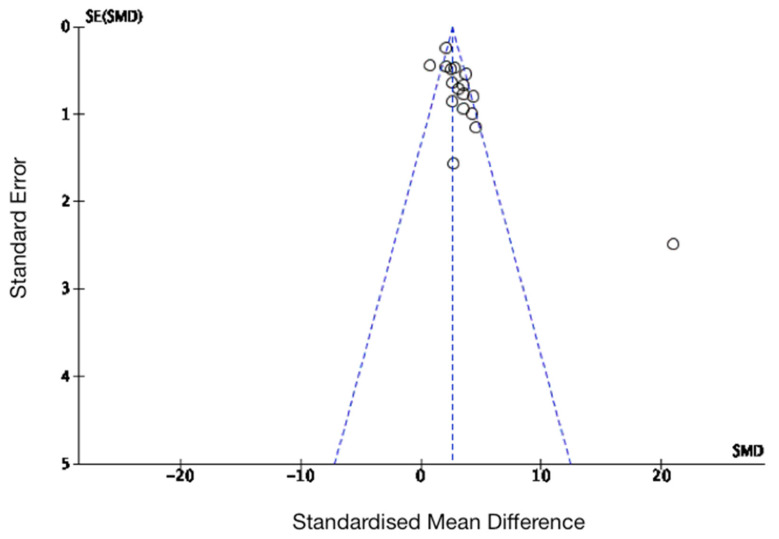
Assessment for publication bias of functional outcomes (Basso–Beattie–Bresnahan (BBB) scores) in rat studies. Standardised mean differences were plotted against standard error; as shown in the funnel plot, there was some evidence of potential bias due an asymmetrical distribution.

**Table 1 biology-10-00230-t001:** Human studies of MSC transplantation for treatment of spinal cord injury. The studies that were selected for systematic review and meta-analysis following literature searches in PubMed, Clinicaltrials.gov, and the Cochrane Library. Abbreviations: MSC, mesenchymal stem cell; C, controlled study; UC, uncontrolled study; R, randomised study; NonRand, nonrandomised study; IT, intrathecal; IV, intravenous; LP, lumbar puncture; Au, autologous MSCs; Allo, allogeneic MSCs; BM, bone marrow-derived MSCs; AT, adipose tissue-derived MSCs; UC, umbilical cord-derived MSCs; MS, multiple sites; CP, cryopreservation; NS, not specified; DMSO, dimethyl sulfoxide.

Author	Study Design	MSC Delivery	Cell Source	MSC Dosage	CP	Participants (*n*)
Mendonca et al., 2014 [61]	UC, NonRand	IT	Au, BM	1 × 10^7^	None	12
Vaquero et al., 2018 [62]	UC, NonRand	IT	Au, BM	3 × 10^8^	10% DMSO	11
Oh et al., 2016 [63]	UC, NonRand	IT	Au, BM	4.8 × 10^7^	None	16
Hur et al., 2016 [15]	UC, NonRand	IT	Au, AT	9 × 10^7^	None	14
Vaquero et al., 2017 [64]	UC, NonRand	IT	Au, BM	1.2 × 10^8^	NS	10
Cheng et al., 2014 [65]	C, Rand	IT	Allo, UC	4 × 10^7^	None	10 treated 24 control
Dai et al., 2013 [66]	C, Rand	IT	Au, BM	8 × 10^5^ (MS)	None	20 treated 20 control
Karamouzian et al., 2012 [14]	C, NonRand	LP	Au, BM	0.7–1.2 × 10^6^	None	11
Ra et al., 2011 [67]	UC, NonRand	IV	Au, AT	4 × 10^8^	None	8
Pal et al., 2009 [68]	UC, NonRand	LP	Au, BM	1 × 10^6^/kg	NS	30
Vaquero et al., 2016 [69]	UC, NonRand	IT	Au, BM	1.3–2.6 × 10^8^	NS	12
El-Kheir et al., 2014 [70]	C, Rand	LP	Au, BM	2 × 10^6^/kg	None	50 treated 20 control
Phedy et al., 2019 [71]	Case study	IT + IV	Au, BM	69.5 × 10^6^	NS	1
Bydon et al., 2020 [72]	Case study	LP	Au, AT	1 × 10^8^	NS	1
Larocca et al., 2017 [73]	Pilot, UC, NonRand	IT	Au, BM	2 × 10^7^	None	5
Chotivichit et al., 2015 [74]	Pilot, UC, NonRand	IT	Au, BM	3 × 10^7^	None	1
Hua et al., 2016 [75]	Pilot, UC, NonRand	IT	Au, UC	4 × 10^7^	None	1
Jarocha et al., 2015 [76]	Case study	IT + IV	Au, BM	1.54 × 10^8^	None	1
Park et al., 2012 [77]	UC, NonRand	LP	Au, BM	1.48 × 10^8^	10% DMSO	10

**Table 2 biology-10-00230-t002:** Animal studies of MSC transplantation for treatment of spinal cord injury. Animal Abbreviations: MSC, mesenchymal stem cell; Allo, allogeneic MSC; BM, bone marrow-derived MSC; AT, adipose tissue-derived MSC; CP, cryopreservation; DMSO, dimethyl sulfoxide.

Author	Species	SCI Phase	MSC Delivery	Cell Source	MSC Dosage	CP	Participants (*n*/Group)
Ban et al., 2011 [78]	Rat	Sub-acute	IT	Allo, BM	1.2 × 10^3^/kg	None	20
Chen et al., 2014 [79]	Rat	Sub-acute	IV	Allo, BM	1 × 10^6^	None	6
Galhom et al., 2018 [80]	Rat	Acute	IT	Allo, BM	1.5 × 10^6^	None	20
Hosseini et al., 2018 [81]	Rat	Acute	IV	Allo, BM	2 × 10^6^/kg	None	15
Karaoz et al., 2012 [82]	Rat	Acute	IT	Allo, BM	3 × 10^5^	None	3
Kim et al., 2013 [83]	Rat	Chronic	IT	Allo, BM	1 × 10^6^	None	12
	Rat	Chronic	IV	Allo, BM	1 × 10^6^	None	12
Kim et al., 2016 [84]	Rat	Sub-acute	IT	Allo, BM	1 × 10^6^	None	12
Mitsuhara et al., 2013 [85]	Rat	Chronic	IV	Allo, BM	3 × 10^5^	None	11
Morita et al., 2016 [59]	Rat	Acute	IV	Allo, BM	1 × 10^6^	None	8
Nakajima et al., 2012 [23]	Rat	Acute	IT	Allo, BM	1 × 10^6^	None	10
Osaka et al., 2010 [58]	Rat	Acute	IV	Allo, BM	1 × 10^6^	None	7
	Rat	Chronic	IV	Allo, BM	1 × 10^6^	None	7
Quertainmont et al., 2012 [86]	Rat	Sub-acute	IV	Allo, BM	1 × 10^6^	None	20
Rosado et al., 2017 [87]	Rat	Acute	IV	Allo, AT	1 × 10^6^	10% DMSO	50
Sandner et al., 2015 [88]	Rat	Acute	IT	Allo, BM	5 × 10^5^	None	12
Torres-Espin et al., 2014 [89]	Rat	Sub-acute	IT	Allo, BM	4.5 × 10^5^	None	7
Wang et al., 2014 [90]	Rat	Sub-acute	IT	Allo, BM	1 × 10^6^	None	16
Wang et al., 2018 [36]	Rat	Acute	IT	Allo, BM	1 × 10^6^	None	15
Lin et al., 2013 [38]	Rabbit	Acute	IT	Allo, BM	5 × 10^6^	None	24
de Almeida et al., 2015 [91]	Mouse	Chronic	IT	Allo, BM	8 × 10^5^	None	8
Neirinckx et al., 2015 [92]	Mouse	Acute	IT	Allo, BM	3 × 10^4^	None	6
Takahashi et al., 2018 [37]	Mouse	Acute	IT	Allo, AT	1 × 10^5^	None	10
Watanabe et al., 2015 [28]	Mouse	Sub-acute	IT	Allo, BM	2 × 10^5^	None	36
Khan et al., 2019 [49]	Dog	Acute	IV	Allo, AT	1 × 10^7^	10% DMSO	4
Lee et al., 2015 [93]	Dog	Sub-acute	IT	Allo, AT	1 × 10^7^	None	4
Lee et al., 2017 [94]	Dog	Intermediate	IT	Allo, AT	1 × 10^7^	None	4

## Data Availability

Not applicable.

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
