# Peer review of "The Comparative Effects of Mesenchymal Stem Cell Transplantation Therapy for Spinal Cord Injury in Humans and Animal Models: A Systematic Review and Meta-Analysis"

_biology, 2021, doi:10.3390/biology10030230_

Round 1

Reviewer 1 Report

My questions have been properly addressed. No further remarks.

Author Response

We are very grateful to the reviewer for their careful consideration of the manuscript and for the recommendation to publish the paper. We hope it is a worthwhile addition to the field.

Reviewer 2 Report

The authors have adequately addressed my previous comments, and I have no further comments to be addressed.

Author Response

(The authors gave the same response as above.)

Reviewer 3 Report

This manuscript is very well-written and presents an excellent comprehensive review of mesenchymal stem cell transplantation for the repair of spinal cord injury. The comparative analysis on animal models and humans is really insightful and would attract broad spectrum of readers. I would recommend this article for publication to Biology journal. I have very few minor suggestions.

  1. The figures especially the charts are very difficult to visualize to achieve any discernible information. A clearer axis labels would help the readers.
  2. I would highly recommend authors to include a figure or a cartoon to demonstrate different type of transplantation technique (IT, IV, LP)
  3. What are the differences in exclusion criteria in step 3 and step 4 on the flow diagram?
  4. For animal models, especially rodents, did any of the referenced articles take immune condition of animals into consideration? If so, did it fall into the exclusion criteria for this study.

Author Response

This manuscript is a resubmission of an earlier submission. The following is a list of the peer review reports and author responses from that submission.

Round 1

Reviewer 1 Report

Among experimental options for SCI repair the transplantations of iPSC, Neural Stem Cells, and bulbar olfactory ensheathing cells might be mentioned here. Immunomodulatory or immunosuppressive properties of the MSCs can ameliorate the phenomenon of sterile neuroinflammation in SCI. The mechanism of MSCs (possible) activity in SCI has not been sufficiently elucidated. Some papers published after April 2020 suggest that only few MSC migrate to the lesion site (10.1016/j.yexcr.2020.112184 ), and such observations suggest indirect effects.

The safety issues and reliability of MSCs have been sufficiently emphasized, but this concern remains valid. The risk of MSC therapy is not determined and definitely not documented, but seems lower than in other stem cell therapies (see Bauer et al, 10.1002/sctm.17-0282 ). The therapy can be associated with infection, fever, and the risk of tumorigenicity is possible (Miura M, Stem Cells, 2006, plus different papers on MSCs connection with sarcomas).

I have got few issues that need addressing:

  1. The manuscript does not contain the information on the amount of transplanted MSC (per kg) or on the type and amount of cryoprotectant (DMSO?) in the analysed studies. DMSO might promote astrogenesis or have neurotoxic (or neuroprotective – conflicting data) properties, and must come under scrutiny.
  2. The difference between MSC administration via lumbar puncture and intrathecal delivery should be explained.

Reviewer 2 Report

In the manuscript entitled "The comparative effects of mesenchymal stem cell transplantation therapy for spinal cord injury in humans and animal models: a systematic review and meta-analysis" the authors perform an extensive review on recent literature on MSC transplantation in spinal cord injury drawing conclusions and or disparities between animal studies and human trials.  It is a well-written review with clear parameters analysed in their study.  The authors have assembled an insightful discussions on animal models of SCI, as well as comparisons of commonly used motor function tests (for either animals or humans).

I have a few questions/comments for the authors:

  • In section 1.2 (lines 92-95), the authors mention the use of MP to treat chronic SCI, suggesting that side effects of MP are a reason to not administer MP.  This is true, but I would encourage the authors to expand on this slightly to include evidence to suggest (aside from side effects), MP has been shown to have no effect on improvement when not delivered acutely after injury. 
  • At the end of section 1.2, the authors mention the use of ChABC (with and without cell transplantation) in SCI stating that it is 'unclear whether such experimental approaches will successfully translate to an effective clinical treatment procedure in humans."  I'm not clear on what the authors are saying here.  If it is the use of ChABC in SCI, there is tremendous pre-clinical evidence to state this is likely to be successful in humans (and ChABC is close to clinical trials now).  If it is the use of cell transplantation in humans, according to the list provided in the paragraph, this may need revision as some of these cell types have been administered in humans (for PD treatment, for brachial plexus treatment, etc).  The aim of this statement is not clear as is.
  • In the methods section, under Exclusion criteria, the authors only include studies on animals with IT, IV, or LP injections.  This is a vital point, and is very critical when comparing to human studies.  The authors should consider discussing this point in detail in the discussion section as so many studies in MSC transplantation as well as others utilise injection into the spinal cord tissue itself.  The authors could discuss here the reasons against that approach.

Minor points:

  • Line 45 and 64, removal 'neuronal' from 'neuronal axons'.  'Axons' is the accepted terminology here.
  • Line 196, edit to 'over time' rather than 'over times'
  • Results (Lines 310-317); some of the statements are repetitive and can be removed 'no acute/subacute cases....' 
  • Results (Figure 3), the authors should consider removing 'acute' and 'subacute' from the graphs since these are not needed for this analysis.

Reviewer 3 Report

First, I really appreciate your effort to search and review those various articles and to perform a meta-analysis.

I also agree with the authors in the context of the clinical importance of SCI and the future value of MSC treatment on SCI.

However, I could not find any differential point or novel finding in current meta-analysis in comparison with recent similar studies that performed meta-analysis of MSC trials on SCI (Xu P and Yang X. Cell Transplant 2019; Muthu S, et al. Cytotherapy 2020).

Yes I understand that the authors suggested the comparison of human and animal trials  as a differential point of their study as mentioned in the title.

However, the study consists of two parallel meta-analyses of human and animal trials of MSC on SCI, but no result from the analysis for direct comparison of human and animal trials. 

Anyway, I am adding some recommendations to improve the readability of your manuscript for further submission.

First, clear description of study background to justify the conduct of current research and study aim is necessary.

You would better radically lessen the general description to introduce subject disease and investigational product. and clearly describe the aim of the study in the introduction section.

Second, you need to describe which statistical techniques were used for the meta-analysis specifically. Your current manuscript seems only to contain the information regarding the tools you used for the meta-analysis.

Third, I think you'd better include the result of the analysis for checking the publication bias.
